# Multiphoton Microscopy Reveals DAPK1-Dependent Extracellular Matrix Remodeling in a Chorioallantoic Membrane (CAM) Model

**DOI:** 10.3390/cancers14102364

**Published:** 2022-05-10

**Authors:** Philipp Kunze, Lucas Kreiss, Vendula Novosadová, Adriana V. Roehe, Sara Steinmann, Jan Prochazka, Carol I. Geppert, Arndt Hartmann, Sebastian Schürmann, Oliver Friedrich, Regine Schneider-Stock

**Affiliations:** 1Experimental Tumor Pathology, Institute of Pathology, University Hospital Erlangen, Friedrich-Alexander University Erlangen-Nürnberg (FAU), 91054 Erlangen, Germany; philipp.kunze@uk-erlangen.de (P.K.); sara.steinmann@klinik.uni-regensburg.de (S.S.); 2Institute of Medical Biotechnology, Friedrich-Alexander University Erlangen-Nürnberg (FAU), Paul-Gordan-Str. 3, 91052 Erlangen, Germany; lucas.kreiss@fau.de (L.K.); sebastian.schuermann@fau.de (S.S.); 3Graduate School in Advanced Optical Technologies (SAOT), Friedrich-Alexander University Erlangen-Nürnberg (FAU), Paul-Gordan-Str. 7, 91052 Erlangen, Germany; 4Clinics of Medicine I, University Hospital Erlangen, Friedrich-Alexander University Erlangen-Nürnberg (FAU), 91054 Erlangen, Germany; 5Czech Center for Phenogenomics, Institute of Molecular Genetics of the ASCR, 142 20 Prague, Czech Republic; vendula.novosadova@img.cas.cz (V.N.); jan.prochazka@img.cas.cz (J.P.); 6Department of Pathology and Legal Medicine, Federal University of Health Sciences of Porto Alegre, Porto Alegre 90050-170, Brazil; aroehe@gmail.com; 7Institute of Pathology, University Hospital Erlangen, Friedrich-Alexander University Erlangen-Nürnberg (FAU), 91054 Erlangen, Germany; carol.geppert@uk-erlangen.de (C.I.G.); arndt.hartmann@uk-erlangen.de (A.H.); 8School of Mechanical, Medical & Process Engineering, Queensland University of Technology (QUT), Brisbane, QLD 4000, Australia

**Keywords:** colon cancer, ECM remodeling, collagen, MPM, uPAR

## Abstract

**Simple Summary:**

The formation of metastasis is not only intricately orchestrated by cancer cells but is also affected by the surrounding extracellular matrix (ECM). The barrier function of the ECM represents an obstacle that cancer cells have to overcome to disseminate from the primary tumor to form metastasis in distant organs. Here, we demonstrate an approach to studying the remodeling of a collagen-rich ECM by colorectal tumor cells using multiphoton microscopy (MPM). This approach allows the analysis of the invasion front of tumors grown on the CAM in 3D. MPM is superior to conventional histology, which is limited to 2D analysis and needs extensive tissue preparation.

**Abstract:**

Cancer cells facilitate tumor growth by creating favorable tumor micro-environments (TME), altering homeostasis and immune response in the extracellular matrix (ECM) of surrounding tissue. A potential factor that contributes to TME generation and ECM remodeling is the cytoskeleton-associated human death-associated protein kinase 1 (DAPK1). Increased tumor cell motility and de-adhesion (thus, promoting metastasis), as well as upregulated plasminogen-signaling, are shown when functionally analyzing the DAPK1 ko-related proteome. However, the systematic investigation of how tumor cells actively modulate the ECM at the tissue level is experimentally challenging since animal models do not allow direct experimental access while artificial in vitro scaffolds cannot simulate the entire complexity of tissue systems. Here, we used the chorioallantoic membrane (CAM) assay as a natural, collagen-rich tissue model in combination with all-optical experimental access by multiphoton microscopy (MPM) to study the ECM remodeling potential of colorectal tumor cells with and without DAPK1 in situ and even in vivo. This approach demonstrates the suitability of the CAM assay in combination with multiphoton microscopy for studying collagen remodeling during tumor growth. Our results indicate the high ECM remodeling potential of DAPK1 ko tumor cells at the tissue level and support our findings from proteomics.

## 1. Introduction

Metastasis represents the most prominent cause of cancer-related deaths caused by the invasive growth of cancer cells into adjacent tissue [1]. By altering the homeostasis of immune and stromal cells, as well as the extracellular matrix (ECM), tumor cells create a favorable tumor microenvironment (TME) [2]. The ECM is a meshwork of proteoglycans and fibrous proteins, mostly fibrillary collagen, which makes up to 90% of the protein content of connective tissue. The three-dimensional (3D) architecture of the collagen network influences cancer cell growth, migration, immune response, and cancer progression [3]. Moreover, certain tumors express matrix remodeling enzymes, such as matrix metalloproteases (MMPs) that can degrade the collagen network and thereby facilitate tumor growth and cancer cell migration [4].

In vitro models are conventionally used to study ECM properties and their functional effects [5]. These models require complex 3D matrices (e.g., hydrogel scaffolds or decellularized tissues) and provide insight into the interplay between physical ECM features and molecular signaling in cancer cells [6]. However, most state-of-the-art models can only assess a limited number of ECM characteristics at a time and, therefore, fail to reflect the full complexity of tissue under in vivo conditions. Most animal models suitable for studying the ECM remodeling potential of cancer cells are costly and time-consuming. In this regard, the in vivo chicken chorioallantoic membrane (CAM) assay represents a promising alternative [7]. Since the generation of 3D matrices as well as artificial collagen fibers is experimentally challenging, the collagen-rich CAM seems to be a suitable model for analyzing the crosstalk between tumor cells and the surrounding ECM [5].

The CAM is fully developed after 10 days of embryonic development and displays a 100–200 µm thick, highly vascularized, and immune-deprived membrane [7]. Cancer cells embedded in Matrigel can be transplanted onto the CAM and develop a tumor within a few days, and tumor angiogenesis, invasion, and metastasis can be monitored [7]. Moreover, the CAM is enriched in fibrillary collagen and resembles connective tissue architecture under physiological conditions. The CAM model can be addressed by a number of imaging techniques [8]. Multiphoton microscopy (MPM) allows for the visualization of vascular structures and ECM components of the CAM, as it enables the label-free imaging of fibrillary collagen fibers by exploiting second harmonic generation (SHG) [9,10]. Beyond SHG, fluorescence signals from exogenous markers or native proteins can be visualized [9].

In contrast to other imaging modalities, multiphoton microscopy achieves larger optical penetration depths and is particularly interesting for deep tissue morphology from native tissues, artificial dyes, or from optically cleared organs [11,12,13]. By exploiting the natural contrast mechanism of SHG, MPM allows in situ imaging without manual sectioning or tissue preparation. In addition, the SHG signal is highly specific to collagen fibers and thus shows great promise for applications in the CAM model to evaluate the ECM remodeling potential of tumor cells with specific gene knockouts. The ECM-altering potential of a human breast cancer cell line was already investigated using SHG microscopy in cell culture and 2D histological sections from animal tissue biopsies [14]. As recently published, the combination of the CAM model and MPM allowed for the visualization of tumor growth into a collagen-rich ECM for colorectal tumor cells with death-associated protein kinase 1 (DAPK1) gene knockout [15]. DAPK1 is a Ca^2+^/calmodulin regulated actin-filament-associated serine/threonine kinase [16]. It is involved in a variety of cellular processes, such as apoptosis, integrin-mediated cell adhesion and migration, and TME modulation [16]. Further, DAPK1 acts as a metastasis suppressor, and HCT116 DAPK1 ko cells showed increased interaction with collagens, especially fibrillary collagen-I, -II, and -IV [15,16,17]. However, a systematic and quantitative analysis applying MPM for the investigation of the ECM remodeling potential of DAPK1 has not yet been shown. By exploiting the natural contrast mechanism of SHG, MPM allowed access to 3D fibrillary collagen structures in situ without manual sectioning or tissue preparation. Beyond the technological refinement of multiphoton imaging, we here also show how the loss of DAPK1 results in a proteome and, in particular, the up-regulation of uPAR, contributing to the observed enhanced ECM remodeling and invasion capabilities.

## 2. Materials and Methods

### 2.1. Cell Lines and Cell Culture

We used the human colorectal tumor cell lines HCT116, the HCT116-derived monoclonal CRISPR/Cas9-generated DAPK1 ko clone 21/9, and SW480. HCT116 and SW480 cells were obtained from ATCC, whereas the DAPK1 ko clone was generated by the genome-editing of HCT116 cells using the CRISPR/Cas9 technology, as recently published [15]. HCT116 WT cells, the tested DAPK1 ko clone, as well as the SW480 cell line were maintained in RPMI 1640 medium supplemented with 10% (*v*/*v*) fetal bovine serum (FBS), 1% penicillin (100 U/mL), and streptomycin (100 U/mL) (all from PAN Biotech, Aidenbach, Germany) and cultured in a humidified atmosphere of 5% CO_2_ at 37 °C. The genotype of the cell lines was confirmed using the Multiplex Cell Authentication (MCA) by Multiplexion (Heidelberg, Germany). All cell lines were verified to have a mycoplasma-free status.

### 2.2. Western Blot Analysis

The protein concentrations of whole-cell lysates were quantified using DC™ Protein Assay Kit (Bio-Rad Laboratories, Inc., Hercules, CA, USA). Subsequently, 35 µg of proteins were resolved by denaturing SDS-polyacrylamide gel electrophoresis and transferred to nitrocellulose membranes overnight. Membranes were blocked with 5% non-fat dried milk in TBST for 1 h at room temperature (RT) and incubated with primary antibodies at 4 °C overnight. After washing the membranes three times with TBST for 5 min they were incubated with HRP-conjugated secondary antibodies for 1 h at RT. Finally, the membranes were washed again three times with TBST, and signals were detected using the Immobilon Western HRP substrate kit (WBKLS0500, Merck Millipore, Burlington, MA, USA) with the GeneGnome imaging system (Syngene, Bengaluru, India).

### 2.3. Chicken CAM Assay

The chicken CAM assay was performed as described previously [15]. Briefly, after opening the eggshell at embryonic developmental day 8, 1 × 10^6^ deep red fluorescently (Deep Red Fluorescence—Cytopainter; ab176736, Abcam, Cambridge, UK) labeled tumor cells per egg were embedded in 40 µL of a 50% Matrigel (Corning^®^ Matrigel^®^ Basement Membrane Matrix phenol-red free, 356237, Corning Inc., Corning, NY, USA) mixture with medium (*v*/*v*) and placed onto the CAM. After five days of incubation, micro-tumors, including the surrounding CAM, were freshly sampled and analyzed. To analyze the ECM remodeling potential upon the loss of DAPK1, the HCT116-derived DAPK1 ko clone 21/9 was grown on the CAM of fertilized chicken eggs.

### 2.4. Histological Staining

Serial sections (3–5 µm) of 4% formalin-fixed and paraffin-embedded CAM xenografts were prepared and stained for hematoxylin-eosin (H&E), uPAR (polyclonal, 1:100, Invitrogen, Waltham, MA, USA), and Sirius red using validated protocols established for clinical routine. Slides with histological staining of the CAM were scanned using a Panoramic MIDI system (camera type: CIS VCC-FC60FR19CL; objective: Plan-Apochromat; magnification: ×40; camera adapter magnification: ×1, 3DHISTECH, Ludwigshafen, Germany) for digital analysis. Collagen polarization and the distribution of CAM xenografts were assessed by Sirius red staining. uPAR staining distribution (cytoplasmic/membranous) and intensity were quantified by a pathologist (CG) and subsequently summarized in an immunohistochemical (IHC) score based on a modified scoring system according to Remmele and Stegner [18]. The IHC score was calculated by multiplying the intensity (0–3) with the abundance in percent (0–100/100). Here, the CAM epithelial layer served as an on-slide internal negative control (intensity score = 0). The intensity scoring scale between 1 and 3 was defined by the mean staining intensity (membranous and cytoplasmic, respectively) within tumor cells throughout the whole cohort, where 1 represented the lowest and 3 the highest positive staining intensity.

### 2.5. Ex Ovo Optical Imaging

Fresh and native CAM samples were imaged by an upright multiphoton microscope system (TriMScope II, LaVision BioTec, Bielefeld, Germany), minutes after extraction. Three-dimensional image stacks were recorded with a 25× water immersion objective using an excitation wavelength of 810 nm. Two-dimensional scans of tissue sections were recorded with a 10× objective as mosaic images. Image stitching was performed using ImageJ macros [19]. Three image channels were recorded for separating SHG from fibrillary collagen (395–415 nm), autofluorescence from NADH (415–485 nm), and the fluorescence signal of the tumor marker (590–650 nm). In the multiphoton microscope system, a mode-locked fs-pulsed Ti:Sa laser (Chameleon Vision II, Coherent, Santa Clara, CA, USA) was used for excitation with an average laser power at the sample of 144 mW, a pulse duration of 150 fs and a repetition rate of 80 MHz. Three-dimensional MPM-image stacks were recorded with a 25× water immersion objective (HC Fluortar L 25×/0.95 W VISIR, Leica microsystems, Wetzlar, Germany); 2D MPM mosaic-images of the tissue sections were recorded with a 10× objective (10×, Nikon, Tokyo, Japan) and stitched via a respective imageJ plugin [19]. The back-scattered signal was collected via the same objective and then spectrally separated by two dichroic beam splitters (ZT405 RDC and T495 LPXR, Chroma, Rockingham, VT, USA). The SHG signal was detected at 405 nm (ET405/20, Chroma, VT, USA) and the auto-fluorescence from NADH in metabolically active cells was detected at 450 nm (450/70 BrightLine HC, Semrock Inc., Rochester, NY, USA), while tumor markers were imaged at 620 nm (ET 620/60, Chroma, VT, USA). Finally, the signals in each of the three channels were detected by ultra-sensitive photo-multiplier tubes (PMT) (H 7422-40 LV 5M, Hamamatsu Photonics, Hamamatsu, Japan). For lateral scanning, a galvanometric scanner was used, and the axial motion of the objective allowed axial scanning for the acquisition of 3D image stacks. Furthermore, the sample stage enabled the highly precise motion of the sample, which could be exploited for the digital stitching of virtual images with a larger field of view. The entire device was controlled by the ImSpector Pro software (v 5.0.233.0, LaVision BioTec, Bielefeld, Germany) and images were saved in the bioformat data format (.ome.tif, GNU public license, University of Dundee and Open Microscopy Environment—OME) [20]. In order to enable equal SHG signals from structures of all spatial orientations, a quarter waveplate (500–900 nm, B.Halle GmbH, Berlin, Germany) with an integrated anti-reflection coating (310–1100 nm, B. Halle GmbH, Berlin, Germany) was integrated behind the back-aperture of the objective.

### 2.6. In Ovo Imaging

For the in ovo imaging of the CAM with the embedded tumor, the eggshell was opened as stated above and imaging was directly performed via the opening without the extraction of the CAM tissue. A 16× objective (HC Fluortar L 16× W VISIR, Leica microsystems, Wetzlar, Germany) was used to ensure a sufficient working distance. The filters were chosen as stated above. However, a quarter-wave plate was not included due to the restricted space between egg and objective. Due to this missing wave plate and due to motion artifacts of the living embryo, the systematic quantification of CAM-ECM was not performed. In the future, custom-designed sample holders could allow a more effective use of space below the objective, while software-based image post-processing could reduce motion artifacts to enable the label-free quantification of collagen structures in the CAM of in vivo/in ovo embryos.

### 2.7. Image Analysis/Morphometry

The alignment of collagen fibers in the ECM was manually quantified based on a systematic scoring by two independent experimenters (PK, LK). The number of images with collagen fibers perpendicular or parallel aligned with respect to the interface was counted in each group. Collagen density was quantified using automated image processing as the median pixel intensity of the SHG channel in the ECM region. To examine the directionality of the ECM fibers, images were scored upon visual inspection by two different analysts (PK, LK). First, images without clearly visible collagen fibers were excluded as outliers (n = 4). The remaining images (n = 85) were judged according to whether the individual and separate collagen fibers were perpendicularly aligned against the interface of the inserted pellet or whether they were parallel to this interface (see evaluation criteria in Appendix A). This more general grouping might seem imprecise. However, it was chosen deliberately to ensure a sufficient number of images in each group and to account for the variance of different orientation angles within the same 3D image. In the case of the collagen density analysis, the pixel intensity from a representative 2D plane of the SHG imaging channel was deduced as the density of fibrillary collagen. Therefore, the ECM was manually annotated as a region of interest (ROI). This selection was based on pre-defined criteria: (i) the ROI should include regions of the ECM with clearly visible fibrillary collagen fibers, (ii) the epithelial layer of the pellet should represent the outer border of the ROI, and (iii) dirt particles that are shown as black spots in all three imaging channels should be excluded from the ROI. A visual example is shown in Appendix A. Again, the selection of the final ROI was confirmed by two independent analysts. Finally, the mean and the median pixel intensity within the ROI were quantified as the fibrillary collagen density for the respective image stack. All image processing was performed using the open-source software Fiji (v1.52s, Wayne Rasband, National Institutes of Health, Bethesda, MD, USA) [21].

### 2.8. Protein Digestion

Cell pellets were lysed in 100 mM triethylammonium bicarbonate containing 2% sodium deoxycholate (SDC) and boiled at 95 °C for 5 min. Protein concentration was determined using a BCA protein assay kit (Thermo Scientific, Waltham, MA, USA), and 20 µg of protein per sample was used for mass spectrometry (MS) sample preparation. Cysteins were reduced with a 5 mM final concentration of tris(2-carboxyethyl) phosphine (60 °C for 60 min) and blocked with a 10 mM final concentration of methyl methanethiosulfonate (10 min RT). Samples were digested with trypsin (trypsin/protein ratio 1/30) at 37 °C overnight. After digestion, samples were acidified with trifluoroacetic acid (TFA) to a 1% final concentration. SDC was removed by extraction to ethylacetate and peptides were desalted using in-house-made stage tips packed with C18 disks (Empore, 3M, Maplewood, OH, USA) as described previously [22].

### 2.9. nLC-MS 2° Analysis

Nano reversed-phase columns (EASY-Spray column, 50 cm × 75 µm ID, PepMap C18, 2 µm particles, 100 Å pore size, Thermo Scientific, Waltham, MA, USA) were used for LC/MS analysis. Mobile phase buffer A was composed of water and 0.1% formic acid. Mobile phase B was composed of acetonitrile and 0.1% formic acid. Samples were loaded onto the trap column (C18 PepMap100, 5 μm particle size, 300 μm × 5 mm, Thermo Scientific, Waltham, MA, USA) for 4 min at 18 µL/min loading buffer composed of water, 2% acetonitrile, and 0.1% TFA. Peptides were eluted with mobile phase B gradient from 4% to 35% B in 120 min. Eluting peptide cations were converted to gas-phase ions by electrospray ionization and analyzed on a Thermo Orbitrap Fusion (Q-OT-qIT, Thermo Scientific, Waltham, MA, USA). Survey scans of peptide precursors from 350 to 1400 *m*/*z* were performed in Orbitrap at 120 K resolution (at 200 *m*/*z*) with a 5 × 10^5^ ion count target. Tandem MS was performed by isolation at 1.5 Th with the quadrupole, higher-energy C-trap dissociation (HCD) fragmentation with a normalized collision energy of 30 and rapid-scan MS analysis in the ion trap. The MS2 ion count target was set to 10^4^ and the maximum injection time was 35 ms. Only those precursors with a charge state of 2–6 were sampled for MS2. The dynamic exclusion duration was set to 45 s with a 10 ppm tolerance around the selected precursor and its isotopes. Monoisotopic precursor selection was turned on. The instrument was run in top speed mode with 2 s cycles.

### 2.10. Mass Spectrometric Analysis for Proteomics

For proteomics analysis, 20 µg of protein from each cell lysate was used for MS sample preparation. Samples were digested with trypsine, separated on nano reversed-phase columns, and used for LC/MS analysis. Eluting peptides were analyzed on a Thermo Orbitrap Fusion (Q-OT-qIT, Thermo Scientific, Waltham, MA, USA). All data were analyzed with the MaxQuant software (version 1.6.3.4) and quantified with the label-free algorithm in MaxQuant [23,24]. Data analysis was performed using Perseus 1.6.1.3 software [25]. The false discovery rate (FDR) was set to 1% for both proteins and peptides, and we specified a minimum peptide length of seven amino acids. The Andromeda search engine was used for the MS/MS spectra search against the Human database (downloaded from Uniprot in July 2019, containing 20,444 entries). Enzyme specificity was set as C-terminal to Arg and Lys, also allowing cleavage at proline bonds and a maximum of two missed cleavages. The dithiomethylation of cysteine was selected as a fixed modification, and N- terminal protein acetylation and methionine oxidation as variable modifications. The “match between runs” feature of MaxQuant was used to transfer identifications to other LC-MS/MS runs based on their masses and retention time (maximum deviation 0.7 min), and this was also used in quantification experiments. Missing data were replaced by values taken from a normal distribution with the mean = 16 and standard deviation = 0.5. The mean of 16 corresponds to the minimum value minus 2, which is the quantification limit. Standard deviation = 0.5 represents the average standard deviation for our replicates. We used this setup to stress the difference between low-expressed and non-expressed proteins according to the data processing software (http://coxdocs.org/doku.php?id=perseus:start, accessed on 05 May 2020). The MS proteomics data have been deposited to the ProteomeXchange Consortium via the PRIDE partner repository with the dataset identifier PXD026072 and 10.6019/PXD026072 [26].

### 2.11. Bioinformatics

The presented GO-term association network of significantly dysregulated proteins after the loss of DAPK1 in the HCT116 ko-clone 21/9 was generated as follows. First, a GO-term subset including 9 GO terms related to the interaction with the ECM and cell motility (GO:0030199, GO:0032963, GO:1903053, GO:0048870, GO:0043062, GO:0034330, GO:0007155, GO:0007166, and GO:0009991) was generated. To split the specific GO terms of all significantly dysregulated proteins (*p* < 0.05) (in addition to the manually added DAPK1 protein) into functional groups related to the GO-term subset of interest, the open-source GOnet web application was used (http://tools.dice-database.org/GOnet/, accessed on 24 June 2020) [27]. The following analysis parameters were used: Species: homo sapiens (9606); input proteins: 299 + 1 (DAPK1); GO namespace: biological_process; analysis type: GO-term annotation; GO subset: custom GO terms (n = 9); output type: interactive network. The visual style of the resulting network was then refined using the Cytoscape 3.7.2 software [28].

### 2.12. uPAR ELISA

The whole-cell lysates of HCT116 WT and DAPK1 ko-clone 21/9 cells were analyzed with an uPAR ELISA kit (Human uPAR ELISA Kit; ab246549, Abcam, Cambridge, UK) according to the manufacturer’s instructions. Whole-cell lysates were prepared using urea lysis buffer (4 M urea, 0.5% SDS, 62.5 mM Tris, pH 6.8) supplemented with a 1% protease inhibitor cocktail (Merck Millipore, Darmstadt, Germany) and 1 mM phenylmethylsulfonylfluorid (Roth, Karlsruhe, Germany) and analyzed with a uPAR ELISA kit (Human uPAR ELISA Kit; ab246549, Abcam, Cambridge, UK) according to the manufacturer’s instructions. Briefly, a uPAR standard dilution ranging from 3200 pg/mL to 0 pg/mL was prepared using the included sample diluent NS. Then, the standard dilutions and cell lysate samples were distributed as technical triplicates in the pre-coated 96-well microplate. Afterward, the antibody cocktail was added to the wells and the sealed plate was incubated for 1 h on a plate shaker. The wells were then washed three times with 1× wash buffer PT prior to the incubation with TMB development solution for 10 min on a plate shaker. The reaction was stopped by adding a stop solution and shaking the plate for 1 min. Finally, the OD was recorded at 450 nm with a multi-label plate reader (Victor™ X3, Perkin Elmer, Rodgau, Germany). Presented data were calculated relative to the uPAR concentration found in the HCT116 WT cells.

### 2.13. Transient uPAR siRNA Transfection

Cells were treated with a pool of siRNAs for uPAR (cat.no.: L-006388-00-0005, Dharmacon, Lafayette, CO, USA) and a non-targeting control (scr; cat.no.: D-001810-10-05, Dharmacon). For this, cells were seeded in 6-well plates (Western blot analysis, 3.5 × 10^5^ per well) or a 10 cm cell culture dish (CAM Assay, 10^6^ cells per plate) and allowed to adhere overnight. Cells were treated the next day with a dilution of Lipofectamine RNAiMAX (13778075, ThermoFischer, Waltham, MA, USA) and the siRNA/non-targeting control in Opti-MEM serum-reduced medium (31985062, ThermoFisher, Waltham, MA, USA) in a 1:1 ratio. This mixture (final concentration: 100 pmol/mL) was then incubated for 20 min at RT and added dropwise to the cells. The cells were cultured with the mixture for 48 h. The uPAR knockdown was assessed by Western blot analysis.

### 2.14. D-Tumor Spheroid-Based Invasion Assay

A 3D-tumor spheroid-based invasion assay was performed as previously described [15]. Briefly, 10^3^/200 µL cells were seeded into each well of a 96-Well ULA round-bottom plate. At the same time, 25 µL of siRNA/scrRNA-Lipofectamine mixture was added to each well (final concentration: 100 pmol/mL). Cells were incubated until spheroid formation for 72 h (DAPK1 ko cells) and 168 h (SW480 cells), respectively. Then, 3D-tumor spheroids were embedded in growth-factor-reduced Matrigel^®^ (356238, Corning) and centrifuged at 300× *g* for 3 min at 4 °C to ensure that the spheroids were in a central position. After incubating the plate for 1 h at 37 °C until the Matrigel had polymerized, 100 µL of complete growth medium was added to each well. Afterward, pictures were taken with an inverted light microscope in the bright field channel at ×4 magnification to document the time point 0 h of the spheroid invasion. The 3D-tumor spheroids were then incubated at 37 °C and 5% CO_2_ for 72 h. After 72 h, pictures were taken again as described before.

### 2.15. Statistical Analysis

All statistical analyses were performed using GraphPad Prism Version 8.00 for Windows. Statistical significance was calculated by 1-way ANOVA, 2-way ANOVA, or Mann–Whitney test. *p*-values of * *p* < 0.05, ** *p* < 0.01 and *** *p* < 0.001 were considered as statistically significant.

## 3. Results

In this study, we investigated the ECM remodeling potential of DAPK1 in colorectal tumor cells. We used proteomic analysis to identify proteins that are dysregulated in DAPK1 ko tumor cells (Figure 1). To further investigate the direct ECM remodeling potential of DAPK1 ko tumors at the tissue level, tumor cells were embedded in an in vivo CAM tissue model and later imaged using MPM (Figure 1). By means of label-free SHG, orientation, as well as the relative density of collagen fibers in the ECM, were analyzed and compared to WT tumors and negative Matrigel controls.

### 3.1. Identification of ECM Remodelers in a Proteomic Analysis

In a mass spectrometry (MS) analysis, 299 proteins were found to be significantly dysregulated after DAPK1 depletion in HCT116 WT cells. These proteins were annotated using a Gene Ontology (GO) term subset. Next, a GO network was generated (Figure 2A). Hereby, 70 proteins (23% of all significantly dysregulated proteins, Appendix A) were associated with biological processes related to ECM interactions (Figure 2B). The enhanced migratory and invasive potential of DAPK1 ko cells might be linked to the up-regulation of proteins associated with cell motility, such as PAK1 (log_2_(FC) = 4.4) and CD47 (log_2_(FC) = 3.3). In fact, PAK1 and CD47 are established prognostic markers related to a worse prognosis in CRC [29,30]. The reduced expression of cell adhesion proteins, such as SORBS3 (log_2_(FC) = −5.1), MCAM (log_2_(FC) = −6.0), or NECTIN3 (log_2_(FC) = −1.2), might result in de-adhesion and disturbed cell junction formation, thereby facilitating the dissemination of tumor cells from the tumor mass during the formation of metastases [31,32,33]. Moreover, we identified an important role of the plasminogen-signaling cascade in DAPK1 ko cells (Figure 2C) and validated the up-regulation of the urokinase plasminogen-activating receptor (uPAR) (log_2_(FC) = 4.0) in DAPK1 ko cells using an ELISA assay (Figure 2D).

### 3.2. Tumor Morphometry Using Conventional Histology and Multiphoton Microscopy

To examine an uPAR-dependent ECM remodeling, we first focused on formalin-fixed in paraffin-embedded (FFPE) tissue slices. In H&E stained histological images, we show that DAPK1 ko tumor cells formed loosely-packed tumor masses that disseminate into the CAM with intense tumor budding of single cells or small cell clusters (Figure 3A). To demonstrate collagen density and fiber distribution, we first used classical Sirius red staining on histological sections from CAM xenografts (Figure 3B). Although collagen-staining intensity appeared lower in the surrounding ECM of DAPK1 ko cells, it was not feasible to quantify the amount and differences in the fiber structure. However, when unstained histological slices were imaged using an established MPM procedure, the fibrillary collagen-specific SHG signal enabled label-free detection (Figure 3C). In both cases, systematic alterations in the collagen network structure related to DAPK1 were not detectable, possibly caused by the high variance within each group as well as by the limitations of a purely two-dimensional (2D) analysis. Nevertheless, we could verify an uPAR dysregulation upon DAPK1 loss in these FFPE CAM tumor slices, as suggested by our proteomic analysis. Here, specific immunostaining for uPAR clearly showed an increased uPAR expression in DAPK1 ko cells in the tumor center, as well as at the invasion front under CAM in vivo conditions (Figure 3D–F). Thus, we suggest that the used classical analysis techniques are rather limited for resolving the uPAR-mediated ECM remodeling.

### 3.3. Quantitative Tumor Morphometry and Image Analysis

In contrast to conventional histology (Figure 3A, B), MPM allows the direct and contact-free imaging of native tissues (Figure 3C). Furthermore, MPM provides molecular imaging contrast without extensive tissue preparations, manual sectioning, and additional staining. By infrared excitation, MPM can further achieve large optical penetration depths, making it ideal for 3D imaging. In order to evaluate the feasibility of this technology for a direct, in vivo investigation of tumor growth on the CAM, we applied 3D multiphoton imaging to the CAM of a living embryo via the open eggshell window using an objective with a large working distance (WD = 8 mm) (Appendix A). It was possible to visualize the invasion front of the tumor (Appendix A). However, the limited space between objective and egg prevented the use of the quarter-wave plate to remove the angular dependence of SHG signal intensity for reliable collagen quantification. Furthermore, image motion artifacts of the living chicken embryo (Appendix A) impeded quantitative analysis. Nevertheless, the ECM remodeling potential of DAPK1 ko tumors was examined from fresh ex vivo CAM biopsies using 3D MPM and morphological post-processing (morphometry). In contrast to bright field microscopy imaging (Figure 4A), this approach allows the direct analysis of tissue morphology on a molecular scale. In order to clearly identify the tumor invasion front during the imaging process, tumor cells were labeled with deep-red fluorescent cell tracking dye prior to embedding in Matrigel. The resulting tumor cell-Matrigel pellets were transplanted onto the CAM of fertilized eggs and sampled after five days of incubation (Figure 1) [15]. Matrigel pellets without any tumor cells served as negative controls.

Conventional bright-field microscopy only visualized blood vessels, whereas MPM revealed fibrillary collagen fibers in the CAM, the natural auto-fluorescence of NADH, as well as the specific red fluorescence tumor marker (Figure 4A). At least six samples per group were investigated and five to six 3D image stacks at different sites at the invasion front were recorded from each sample. As shown in Figure 4B, the tumor invasion front was clearly defined by the border between fibrillary collagen (blue) and the labeled tumor cells (red). While fibrillary collagen fibers in native CAM were mainly characterized by unorganized networks of mesh fibers, individual collagen fibers were clearly detectable in the ECM at the invasion front or at the interface between Matrigel pellet and CAM, which might be related to growth factors in medium/Matrigel (Appendix A). Here, the simple embedding of a Matrigel pellet onto the CAM was sufficient to generate a highly functional microenvironment with a relevant ECM structure anisotropy under in vivo conditions. Moreover, we observed that the directionality of collagen fibers showed apparent differences between the WT, DAPK1 ko, and Matrigel pellets. The fraction of images with collagen fibers aligned perpendicularly towards the plain Matrigel pellet in the surrounding CAM was significantly higher compared to WT or DAPK1 ko tumors, while the transplantation of HCT116 WT cells resulted in the opposite effect with the majority of collagen fibers running in parallel to the invasion front (Figure 4B). This phenomenon might be explained by the uniform growth of the tumor cells, which is thought to push the fibers aside, resulting in a reorientation of fibrillary collagen fibers compared to the tumor cell-free Matrigel samples (Figure 4B) [15]. On the other hand, DAPK1 ko tumors had larger areas with perpendicular fiber alignment compared to HCT116 WT tumors, thereby resembling the fiber orientation found in the Matrigel controls (Figure 4B).

Subsequently, images were quantitatively judged by a fibrillary collagen fiber alignment score with respect to the invasion front. The quantification confirmed the initial observations and showed a significant increase in perpendicular fibers in the DAPK1 ko group (Figure 4C). This supports the hypothesis that the loss of DAPK1 in HCT116 cells increases the ECM remodeling potential of this already aggressive tumor cell line. To confirm this hypothesis, the fibrillary collagen density was quantified using semi-automated image processing (quantitative morphometry). After selecting a representative 2D plane from each 3D image stack, the ECM area was manually annotated following pre-defined criteria (see Appendix A) by two independent experimenters. Collagen density was determined as median SHG pixel intensity within the selected region of interest (ROI). Fibrillary collagen density was reduced in DAPK1 ko samples, suggesting the active degradation of fibrillary collagen in the ECM. However, no statistical significance was reached due to the high variance within the WT group (Figure 4D). Active ECM degradation seems to enable the tumor to resolve the barrier function of the ECM, thus promoting the movement of tumor cells towards blood vessels.

### 3.4. Influence of uPAR on ECM Remodelling during Tumor Cell Invasion in DAPK1 ko Cells

Next, we analyzed how the up-regulation of uPAR in DAPK1 deficient HCT116 cells enhanced tumor cell invasion by enabling the remodeling of the surrounding ECM. For this, uPAR expression was successfully silenced by RNAi-mediated uPAR knockdown in HCT116 DAPK1 ko cells (Figure 4E). Subsequently, the ECM remodeling capacity of these cells was studied on CAM tissue with the same MPM approach. The successful uPAR silencing in the CAM xenografts was demonstrated by the IHC staining of uPAR on the same tumors that were analyzed with MPM (Appendix A). Similar to Matrigel and HCT116 DAPK1 ko samples (Figure 4B,C), scrRNA-treated tumors showed more collagen fibers perpendicular toward the invasion front (Figure 4F,G), and the orientation of uPAR siRNA-treated xenografts was more parallel to the invasion front (Figure 4F,G), as observed in the HCT116 WT group (Figure 4B,C). Hence, the effects on the collagen orientation by the up-regulation of uPAR in HCT116 DAPK1 ko cells were reversible after siRNA transfection. Similarly, the fibrillary collagen density was slightly increased in uPAR siRNA-treated tumors in comparison to the scrRNA group (Figure 4H).

### 3.5. In Vitro Simulation of the Early Steps of the Metastatic Cascade Using 3D-Tumor Cell Invasion Assay

To confirm that the enhanced tumor growth, as well as the increased ECM remodeling potential of HCT116 DAPK1 ko cells in CAM tissue, was caused by a DAPK1-dependent up-regulation of uPAR (Figure 5A), we performed a 3D-tumor cell invasion assay in vitro. Therefore, HCT116 DAPK1 ko cells were transfected with uPAR siRNA and scrRNA, respectively, during the generation of 3D-tumor spheroids and subsequently embedded in Matrigel to simulate the invasion into the ECM. As shown in Figure 5B, the silencing of uPAR significantly reduced the invasiveness of these cells after 72 h of tumor growth (Figure 5C,D). To strengthen our hypothesis, we repeated the experiment with SW480 colon tumor cells (Figure 5E–H). SW480 is a colon cancer cell line, which was generated from a primary tumor that has formed metastasis into the lymph nodes [35]. These SW480 cells showed the same pattern regarding DAPK1 and uPAR expression, as HCT116 DAPK1 ko cells (Figure 5A). Similar to HCT116 DAPK1 ko cells, the knockdown of uPAR in SW480 cells significantly reduced the invasive potential after 72 h (Figure 5F–H). In addition, uPAR siRNA-treated spheroids were very compact with a smooth surface, whereas scrRNA-treated cells formed more loosely packed spheroids with several tumor extensions growing into the Matrigel (Figure 5F). This invasive growth pattern is strongly reminiscent of the enhanced tumor budding observed by DAPK1 ko cells in vivo [15].

## 4. Discussion

It is well known that the geometry, orientation, and density of collagen fibers affect the invasive capability of tumor cells. Moreover, the collagen signature of the associated TME, including collagen density and collagen fiber organization, serves to stage the tumor progression levels in breast cancer [36]. During the early stages of tumor invasion, the TME collagen amount is increased, and collagen fibers align as straight fibers parallel to the tumor border, resulting in contracting forces [36,37]. This creates a more restrictive and challenging 3D environment, reducing cell motility and restraining tumor expansion [38]. At later stages, tumors can overcome this barrier through an active degradation of the collagen ECM by activating MMPs. This generates local conduits beneficial for cell invasion [37]. In addition, mixed forms or collagen fibers oriented along cell invasion directions seem to facilitate the migration of small cell clusters or single cells in comparison to the bulk tumor mass [5]. Although the in vivo or in ovo imaging of live tumor cells on the CAM using immunofluorescence microscopy has already been reported [39], only a few papers demonstrate the systematic evaluation of the ECM. For this purpose, SHG microscopy is an essential tool, since it provides the direct and label-free imaging of the ECM, while also allowing reference staining with specific tumor markers, as shown here. In contrast to previous work on ECM remodeling in the context of the CAM model, we were able to refine the imaging hardware by integrating a quarter-wave plate in the optical path, thereby enabling the robust quantification of the SHG signal from collagen structures [15]. The systematic study design and the SHG quantification shown here allowed an in-depth analysis of the tumor cell–ECM interaction in colon cancer. Hence, we could observe that the presence and growth of HCT116 cells in the CAM tissue resulted in an ECM pattern such as that observed at the early stages of tumor invasion in breast cancer [36]. In contrast, the collagen patterns of DAPK1 ko cells resembled later stages of tumor invasion with more fibers along the invasion direction and reduced collagen density [40]. Furthermore, these data support our previous findings, where DAPK1 ko cells adapted faster to their surrounding environment after being transplanted on brain precision-cut tissue slices and cultivated ex vivo [15]. Our results on DAPK1 indicate how enhanced tumor budding potential and active ECM remodeling might be coupled, thus facilitating this invasive process [5,15].

A better understanding of DAPK1-mediated cellular functions is crucial, as more than 50% of CRC harbor an inactivated DAPK1, mostly by promoter hypermethylation [41]. In a recent paper, DAPK1 loss triggered chemo-resistance and metastasis in CRC via a ZEB1-WNT signaling pathway [17]. Our proteomics analysis added a novel network for DAPK1 loss in relation to the ECM remodeling potential of tumor cells that helps to explain the functional consequences of DAPK1 loss in tumors. Notably, the dysregulation of proteins responsible for the structural organization of the ECM, such as COL6A1 (log_2_(FC) = 4.3), might explain disturbances in responses to extracellular stimuli [34]. Central to the plasminogen-signaling cascade is uPAR, another valuable prognostic marker in CRC, involved in MMP-mediated ECM degradation [34,42]. The activation of cell surface receptor signaling pathways, such as the plasminogen-signaling cascade, showed that the active ECM remodeling and cellular response to extracellular signals are tightly connected. We suggest that the surface receptor uPAR might be a potential candidate to explain the active remodeling of the fibrillary collagen ECM by DAPK1 ko cells on a molecular biological scale [42]. We have already demonstrated this uPAR dysregulation upon DAPK1 loss in a Nanostring gene expression analysis [15]. Interestingly, the highly aggressive metastatic SW480 cells [35] showing the same expression pattern (DAPK1 low, uPAR high) as HCT116 DAPK1 ko cells were less invasive under uPAR-si knockdown in a 3D spheroid assay. This mechanistic study strongly suggested a uPAR-mediated ECM remodeling at the tumor invasion front where we find remarkably reduced DAPK1 levels [15].

## 5. Conclusions

In conclusion, the combination of the CAM assay with advanced MPM imaging technology allowed the direct monitoring of alterations at the tumor invasion front in a feasible and fast manner in real-time without the need for artificial scaffolds. MPM was superior to conventional histology or bright field microscopy, which are usually limited to 2D analysis and require extensive tissue preparation that might alter the homeostatic tissue architecture. Using this approach, it was possible to identify a characteristic ECM remodeling behavior of CRC tumors that lack DAPK1. An in-depth proteomics analysis further complemented the evaluation and provided insights into the molecular and functional consequences of DAPK1 loss. In the future, minimally invasive MPM could be used for the in vivo optical histology of the CAM in a longitudinal study on tumor growth with the repeated imaging of the same specimen at different stages of tumor growth.

## Figures and Tables

**Figure 1 cancers-14-02364-f001:**
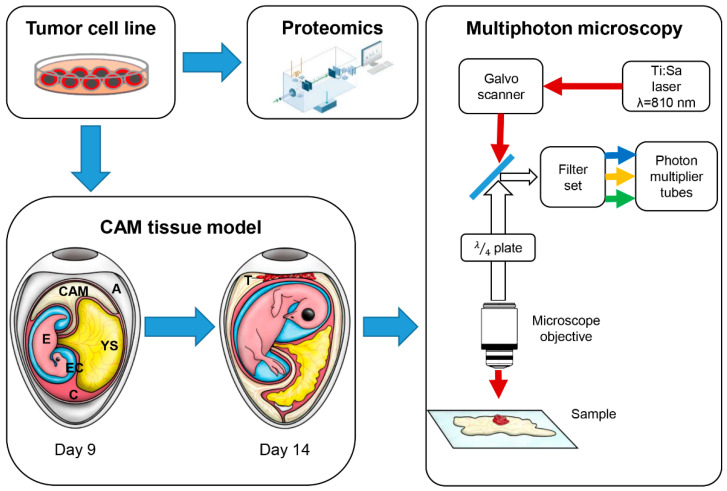
Schematic representation of the experimental study to analyze proteomics as well as the ECM remodeling potential of tumor cells. Tumor cell lines harboring a specific gene ko were investigated by proteomics in order to study the DAPK1-dependent proteomic signature. Tumor cells were embedded onto the CAM of a fertilized hen egg and finally imaged using multiphoton microscopy. First, deep-red fluorescence-labeled and Matrigel-embedded tumor cells were transplanted onto the CAM (CAM model: CAM: chorioallantoic membrane; A: albumen; E: embryo; EC: embryotic cavity; YS: yolk sack; C: chorion; T: tumor). After 5 days, CAM tumors were prepared and subsequently analyzed using an upright multiphoton microscope. A quarter-wave plate was integrated to ensure equal SHG signal intensities across different spatial polarization orientations.

**Figure 2 cancers-14-02364-f002:**
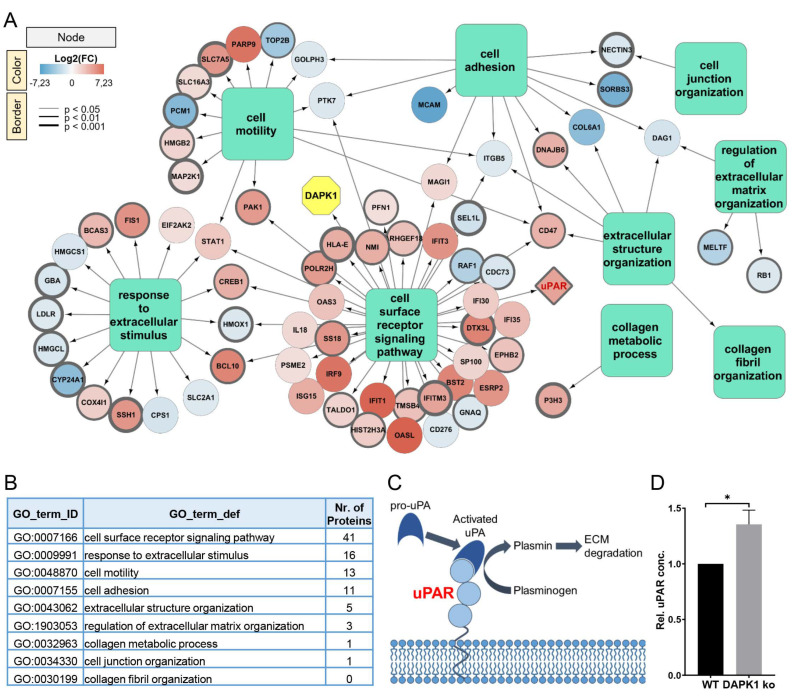
DAPK1-dependent proteomics signature. (**A**) GO-term network of all 70 proteins annotated with a GO-term subset. The node color, ranging from blue to red, represents the log2(FC) value of the respective protein, whereas the thickness of the node border represents the *p*-value. The node for DAPK1 (yellow) was added manually to the analysis. (**B**) GO-term IDs and definitions of the selected GO-term subset including the number of annotated proteins. (**C**) Schematic representation of the role of uPAR in the plasminogen-signaling cascade resulting in ECM degradation (Figure adapted from Brungs et al., 2017 [34], CC-BY for unrestricted use). (**D**) Validation of uPAR up-regulation upon DAPK1 loss in HCT116 WT cells using an ELISA assay (Mann-Whitney test: * *p* < 0.05). Data represented as mean ± SEM.

**Figure 3 cancers-14-02364-f003:**
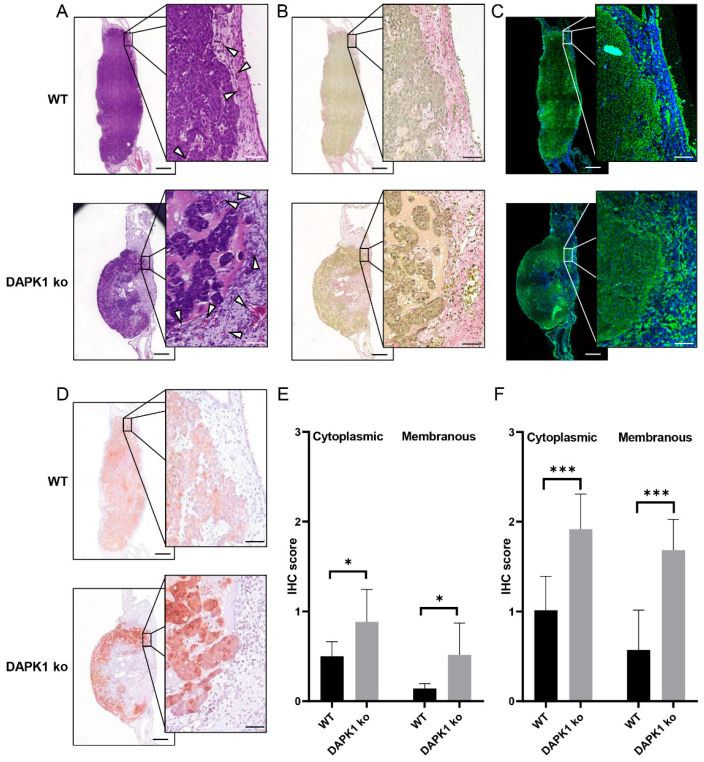
Evaluation of qualitative collagen fiber density and distribution, as well as uPAR expression in histological slices of CAM xenografts in 2D. (**A**) H&E staining of CAM xenografts derived from HCT116 WT cells (above) and DAPK1 ko tumor cells (below). Arrowheads indicate intra- and peri-tumoral chicken vessels. (**B**) Collagen staining of histological sections with Sirius red for the analysis of collagen network structure and density. (**C**) Label-free visualization of fibrillary collagen (blue) and natural auto-fluorescence from NADH (green) of unstained sections using MPM. (**D**) Conventional IHC stainings for the uPAR of CAM xenografts. (**E**) Cytoplasmic and membranous uPAR IHC score of HCT116 WT cells and DAPK1 ko CAM tumors at the invasion front and (**F**) at the tumor center (WT: n_samples_ = 7; DAPK1 ko: n_samples_ = 6; 2-way ANOVA: * *p* < 0.05; *** *p* < 0.001). (Scale bar main image = 400 µm; scale bar zoomed inlet 50 µm). Data represented as mean ± SD.

**Figure 4 cancers-14-02364-f004:**
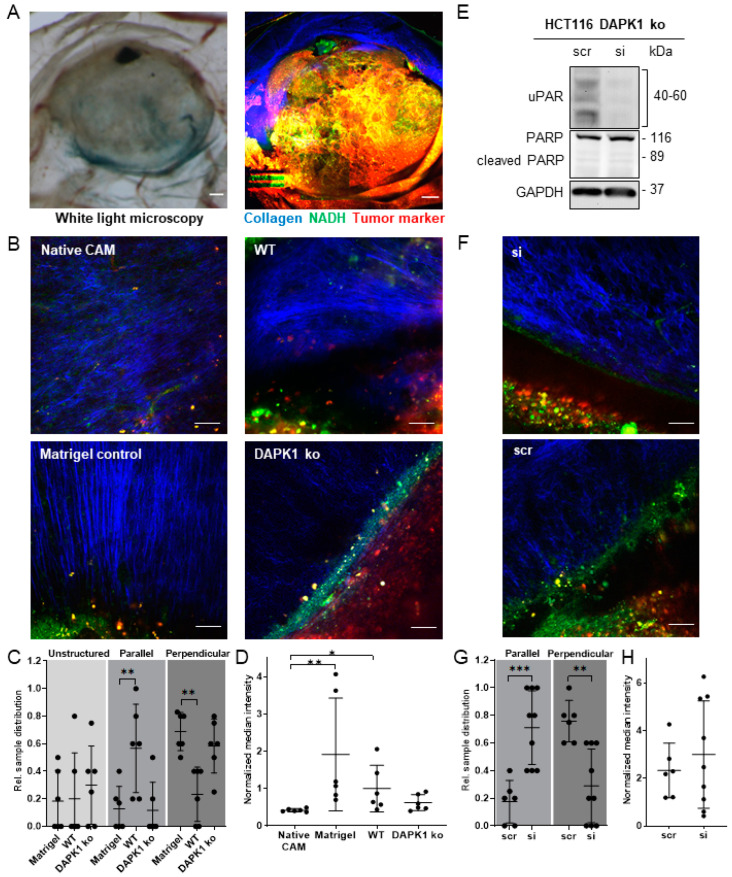
Evaluation of apparent collagen fiber density and distribution in CAM xenografts in 3D using MPM. (**A**) Freshly prepared tumor—including surrounding CAM—tissue imaged using bright-field microscopy and a digitally stitched MPM mosaic (scale bar = 500 µm). The color code shows the targeted molecules for the respective imaging channel. (**B**) Collagen network at the tumor interface under different tested conditions: Native CAM with an unstructured, Matrigel and DAPK1 ko clone with perpendicular, and HCT116 WT with parallel fiber orientation (scale bar = 50 µm). (**C**) Relative fraction of images with unstructured, parallel, or perpendicular collagen fibers’ orientation relative to the invasion front or the Matrigel interface (n_samples_ = 6; 2-way ANOVA: ** *p* < 0.01). (**D**) Collagen density at the Matrigel interface and the invasion front, respectively, expressed by normalized median pixel intensity of the SHG signal within the manually annotated ECM of one representative slice from each 3D stack (n_samples_ = 6; 1-way ANOVA: * *p* < 0.05; ** *p* < 0.01). (**E**) Western blot of DAPK1 ko cells after siRNA and scrRNA transfection, demonstrating the successful knockdown of uPAR without inducing cell death (original Western blot images presented in Appendix A). (**F**) Representative MPM images of siRNA-tumors, resembling HCT116 WT tumors, while scrRNA-derived tumors resemble DAPK1 ko xenografts (scale bar = 50 µm). (**G**) Relative collagen fiber orientation of uPAR siRNA- and scrRNA-treated tumor cells (scr: n_samples_ = 6; si: n_samples_ = 9; 2-way ANOVA: ** *p* < 0.01; *** *p* < 0.001). (**H**) Normalized median pixel intensity of the SHG signal of uPAR siRNA- and scrRNA-treated xenografts (scr: n_samples_ = 6; si: n_samples_ = 9; Mann–Whitney test: n.s.). Data represented as mean ± SD.

**Figure 5 cancers-14-02364-f005:**
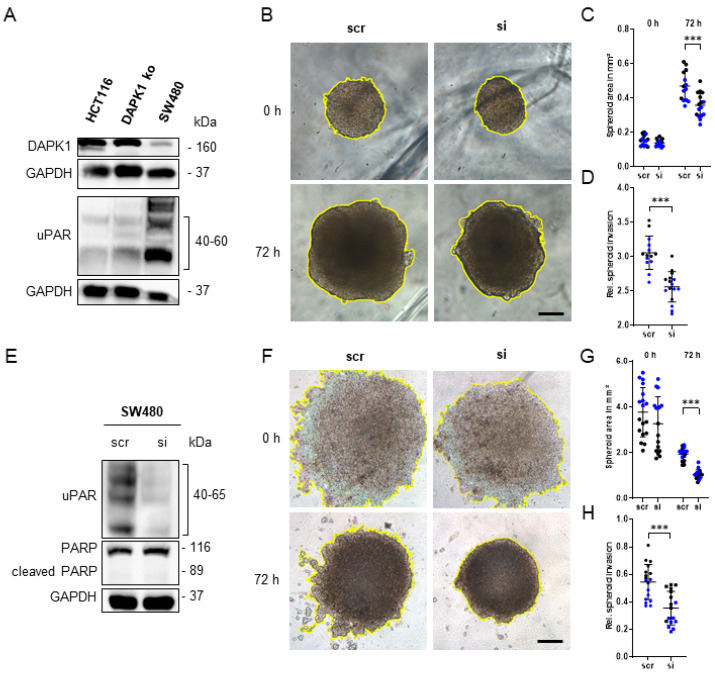
uPAR-dependent tumor cell invasion in vitro. (**A**) Western blot analysis of DAPK1 (lower band at 160 kDa) and uPAR expression in HCT116 WT, DAPK1 ko, and SW480 cells. In contrast to HCT116 WT cells, DAPK1 ko and SW480 cells show the same expression pattern (DAPK1 low, uPAR high) (original Western blot images presented in Appendix A). (**B**) 3D-tumor spheroid-based invasion assay with uPAR siRNA/scrRNA-treated HCT116 DAPK1 ko cells. (**C**) Area quantification of uPAR siRNA/scrRNA-treated HCT116 DAPK1 ko 3D-tumor spheroids after embedding in Matrigel (0 h) and after 72 h in mm^2^ (n_samples_ = 16; 2-way ANOVA: *** *p* < 0.001). (**D**) The 3D-tumor spheroid invasion of uPAR siRNA/scrRNA-treated HCT116 DAPK1 ko cells after 72 h relative to time point 0 h (n_samples_ = 16; Whitney test: *** *p* < 0.001). (**E**) Western blot of SW480 cells after siRNA and scrRNA transfection, demonstrating the successful knockdown of uPAR without inducing cell death (original Western blot images presented in Appendix A). (**F**) A 3D-tumor spheroid-based invasion assay with uPAR siRNA/scrRNA-treated SW480 cells. (**G**) Area quantification of uPAR siRNA/scrRNA-treated HCT116 DAPK1 ko 3D-tumor spheroids after embedding in Matrigel (0 h) and after 72 h in mm^2^ (scr: n_samples_ = 18; si: n_samples_ = 17; 2-way ANOVA: *** *p* < 0.001). (**H**) The 3D-tumor spheroid invasion of uPAR siRNA/scrRNA-treated SW480 cells after 72 h relative to time point 0h (scr: n_samples_ = 18; si: n_samples_ = 17; Whitney test: *** *p* < 0.001). Black dots in graphs represent the first biological replicate and blue dots represent the second biological replicate. Data represented as means ± SD.

## Data Availability

The data is available upon reasonable request to the authors.

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
