# Peer review of "Multiphoton Microscopy Reveals DAPK1-Dependent Extracellular Matrix Remodeling in a Chorioallantoic Membrane (CAM) Model"

_cancers, 2022, doi:10.3390/cancers14102364_

Round 1
Reviewer 1 Report
In your paper you impressively demonstrate the application of MPM in vivo and in situ for ECM remodelling in CAM model.
The title however, does not reveal the whole content, e.g. you also show the upregulation of uPAR related to DAPK1 ko in HCT116 cells and CAM tumors.
If possible please add a subtitle or add DAPL1 and uPAR as keywords
Author Response
Reviewer 1:
In your paper you impressively demonstrate the application of MPM in vivo and in situ for ECM remodelling in CAM model.
- Thank you for this encouraging comment. We are glad that form, style and language are appropriate.
The title however, does not reveal the whole content, e.g. you also show the upregulation of uPAR related to DAPK1 ko in HCT116 cells and CAM tumors.
- We agree and changed the title to “Multiphoton microscopy reveals DAPK1 dependent extracellular matrix remodeling in a chorioallantoic membrane (CAM) model”.
If possible please add a subtitle or add DAPL1 and uPAR as keywords
- We adapted the keywords and added uPAR and removed CAM, as it appears already in the title.
Reviewer 2 Report
The paper explores interesting topic and employs a broad spectrum of methods and collaborative approach to achieve the goals. It brings new, inspirative and promising results both in the field of biophotonics as well as tumor invasion. Introduction provides overview of newest findings in the field, necessary background for the explored topic. Material section is clear and despite very complex approach, still not overwhelming. The research body consisted of several teams, hence the writing of the manuscript is more challenging. Below is list of minor comments and corrections of mistakes, which slipped your attention.
- L92 "The technique…" Put the sentence better in the context or move to different place.
- L135 Please provide reasoning why there was a need to use SPF chicken eggs.
- L155-157 Provide either more information about the IHC score or add the reference. What was used as a negative/positive control?
- Are lines 160-166 simplified version of more detailed description later (L166 onwards)? If yes, it might be confusing and redundant.
- L174 Did you mean Nikon objective?
- L193 In ovo imaging is very nice approach, however, provided information is rather brief. Would cultivation of embryo ex ovo (in a plastic dish) improve accessibility of CAM surface for better imaging?
- L284 Bioinformatics - Part of this paragraph deals with bioinformatics, part with the statistics. This should be reflected in the title. Statistics part, however, is rather slim and doesn't refer specifically what analysis was used on what data (e.g. Fig. 3 e,f).
- When two analyst/scorers were used, how did you check the consistency of their scoring (e.g. coefficient of concordance)?
- L299-300 K-W and M-W tests were used on the same graph (Fig.2d -Validation of uPAR up-regulation upon DAPK1 loss in 386 HCT116 using an ELISA assay)? Since there are two groups, probably M-W was used.
- Fig. 1 Misspelled Protenomics/Proteomics. Figure 1 would probably better fit in m&m section.
Author Response
Reviewer 2:
The paper explores interesting topic and employs a broad spectrum of methods and collaborative approach to achieve the goals. It brings new, inspirative and promising results both in the field of biophotonics as well as tumor invasion. Introduction provides overview of newest findings in the field, necessary background for the explored topic. Material section is clear and despite very complex approach, still not overwhelming. The research body consisted of several teams, hence the writing of the manuscript is more challenging. Below is list of minor comments and corrections of mistakes, which slipped your attention.
- Thank you very much for the time and effort of this supportive, fair and encouraging review!
L92 "The technique…" Put the sentence better in the context or move to different place.
- Done
L135 Please provide reasoning why there was a need to use SPF chicken eggs.
- SPF eggs were used but are not essential for a successful experiment. Hence, we omitted this information in the material&method section, as it could be misleading.
L155-157 Provide either more information about the IHC score or add the reference. What was used as a negative/positive control?
- IHC scoring was based on a modified scoring system according to Remmele and Stegner (1987). The CAM epithelial layer served as an on-slide internal negative control (intensity score = 0). The intensity scoring scale between 1-3 was defined by the mean staining intensity (membranous and cytoplasmic, respectively) within tumor cells throughout the whole cohort, where 1 represented the lowest and 3 the highest positive staining intensity.
Are lines 160-166 simplified version of more detailed description later (L166 onwards)? If yes, it might be confusing and redundant.
- Thanks for this critical comment. The redundant paragraph was removed from the manuscript.
L174 Did you mean Nikon objective?
- Yes, we changed Nicon into Nikon
L193 In ovo imaging is very nice approach, however, provided information is rather brief. Would cultivation of embryo ex ovo (in a plastic dish) improve accessibility of CAM surface for better imaging?
- The ex ovo embryo cultivation might be a suitable approach, however, we do not have the experimental lab prerequisites available to perform such experiments. We added the following paragraph in the material&method section providing more detailed information about in ovo imaging:
“Due to this missing wave plate and due to motion artefacts of the living embryo, systematic quantification of CAM-ECM was not performed. In the future, custom-designed sample holders could allow a more effective use of space below the objective, while soft-ware-based image post-processing could reduce motion artefacts to enable label-free quantification of collagen structure in CAM of in vivo / in ovo embryos.”
L284 Bioinformatics - Part of this paragraph deals with bioinformatics, part with the statistics. This should be reflected in the title. Statistics part, however, is rather slim and doesn't refer specifically what analysis was used on what data (e.g. Fig. 3 e,f).
- The bioinformatics and the statistics part were separated in the material&methods section. We agree with the reviewer that the information must be given to reader for every statistical analysis. Hence, we added in the figure legend the statistical test used for each graph.
When two analyst/scorers were used, how did you check the consistency of their scoring (e.g. coefficient of concordance)?
- First, evaluation criteria were defined by both analysts (see Figure S2). Next, the set of images was evaluated regarding this criteria. Divergent cases were discussed in a back and forth manner in several iterations, until all discrepancies were elucidated and experimentors agreed on how to judge difficult cases. However, these case represented the minority. The advantage of this approach was that both analysist come from different research areas and thus have different expertise, which helped to interpret difficult cases by judging from different points of view.
L299-300 K-W and M-W tests were used on the same graph (Fig.2d -Validation of uPAR up-regulation upon DAPK1 loss in 386 HCT116 using an ELISA assay)? Since there are two groups, probably M-W was used.
- Thank you very much for this detailed comment. Indeed, a M-W test was performed in the graph presented in Figure 2d.
Fig. 1 Misspelled Protenomics/Proteomics. Figure 1 would probably better fit in m&m section.
- Done
This manuscript is a resubmission of an earlier submission. The following is a list of the peer review reports and author responses from that submission.
Round 1
Reviewer 1 Report
Thank you for addressing my suggestions and remarks. The manuscript has improved significantly, however the nature of DAPK1 k.o. remains obscure to me.
In Fig. 5A and the figure legend, as well as in the discussion l. 600 ff. it still is not clear what you show. The WB does not show DAPK1 k.o., nor upregulation of uPAR as you state in the text. Also supplemental figures show the same results.
This makes it hard to believe in your k.o. cells and puts a question mark above all results.
Reviewer 2 Report
This is a resubmission of a study by Kunze et al. that examines the role of the DAPK1 kinase in ECM remodeling and cell invasion of cancer cells using different approaches, including proteomics, immunohistochemistry, and multiphoton microscopy (MPM) in CAM xenografts and 3D-spheroid-based invasion assays.
Even though the authors have responded to most comments, the paper still suffers from several weaknesses.
The central claim of the abstract is the use of MPM to demonstrate the invasive capacity of DAPK1-deficient cells when MPM 1 is only one of the tools used to demonstrate the ECM remodeling induced by DAPK1 knockdown.
The conclusion from the mechanistic aspect of the paper that UPAR is downstream of DAPK1 should be supported by rescue experiments.
Figure 5 should be rearranged to reflect the order of description in the text.
Western blot demonstrating DAPK1 knockdown (Figure 5A) is either not convincing or the band of interest not indicated. Western blots (Figure 5AandB) should be quantified, and statistics provided.
The resolution of Figure 5 is low, and this should be corrected.